# Post-Radiotherapy Exosomal Non-Coding RNA and Hemograms for Early Death Prediction in Patients with Cervical Cancer

**DOI:** 10.3390/ijms25010126

**Published:** 2023-12-21

**Authors:** Oyeon Cho

**Affiliations:** Gynecologic Cancer Center, Department of Radiation Oncology, Ajou University School of Medicine, Suwon 16499, Republic of Korea; oyeoncho@ajou.ac.kr; Tel.: +82-31-219-4195; Fax: +82-31-219-5894

**Keywords:** exosomal non-coding RNAs, blood cell dynamics, early death prediction, cervical cancer, radiotherapy

## Abstract

Concurrent chemo-radiotherapy (CCRT) is linked with accelerated disease progression and early death (ED) in various cancers. This study aimed to assess the association of plasma levels of exosomal non-coding ribonucleic acid (RNA) (ncRNA) and blood cell dynamics with ED prediction in patients with cervical cancer undergoing CCRT. Using propensity score matching, a comparison of complete blood counts (CBCs) was performed among 370 CCRT-treated patients. Differences in ncRNA and messenger RNA (mRNA) expression before and after CCRT in 84 samples from 42 patients (cohort 2) were represented as logarithmic fold change (log_2_FC). Networks were constructed to link the CBCs to the RNAs whose expression correlated with ED. From the key RNAs selected using multiple regression of all RNA combinations in the network, CBC dynamics-associated ncRNAs were functionally characterized using an enrichment analysis. Cohort 1 (120 patients) exhibited a correlation between elevated absolute neutrophil counts (ANC) and ED. Cohort 2 exhibited a prevalence of microRNA (miR)-574-3p and long intergenic non-protein coding (LINC)01003 ncRNA, whose expression correlated with ANC and hemoglobin values, respectively. Conversely, acyl-coenzyme A thioesterase 9 (*ACOT9*) mRNA was relevant to all CBC components. An integrative analysis of post-CCRT ncRNA levels and CBC values revealed that the patients with miR-574-3p-LINC01003-*ACOT9* log_2_FC) < 0 had a better prospect of 30-month disease-specific survival. These findings indicate that miR-574-3p and LINC01003 could serve as ED prognostic biomarkers.

## 1. Introduction

A subset of patients receiving primary concurrent chemo-radiotherapy (CCRT) for cervical cancer exhibit an accelerated disease progression and a mortality rate that surpasses prognostic expectations [1]. This phenomenon of early death (ED) has been documented in various cancer types, including breast, stomach, lung, head and neck, colorectal, and bladder cancers, and it is potentially resistant to the three predominant cancer treatment modalities, namely surgery, chemotherapy, and radiation therapy (RT) [2,3,4,5,6,7,8,9,10,11]. However, the precise etiology underlying this phenomenon remains unknown. One plausible hypothesis posits that cancer progression could potentially accelerate following treatment if the host is unable to effectively manage the inflammatory and stress-inducing conditions associated with cancer, therapy, environmental or dietary factors, and chronic infection [12].

The functions of various non-coding ribonucleic acids (RNAs) (ncRNAs) that mediate inflammation caused by pathogens and environmental causes and the development and progression of cancer have been reported in gastric, colon, cervical, head and neck, lung, and esophageal cancers [13]. This suggests that some ncRNAs may play an important role in regulating the inflammatory responses associated with cancer progression.

Exosomes (measuring approximately 40–160 nm on an average) are extracellular vesicles (EVs) that comprise nucleic acids, proteins, lipids, amino acids, and metabolites. They are secreted by both cancer and normal cells to enable the regulation of intercellular communication [14]. They assume a pivotal role in diverse physiological processes, such as intracellular signal transmission, and they engage in activities including cancer progression, immune response modulation, inflammation, and stress response regulation [15,16]. ncRNAs present in exosomes are anticipated to have a crucial role in modulating inflammatory responses by regulating diverse messenger RNAs (mRNAs) [17]. The abundance of normal cell-derived exosomes in the blood facilitates the identification of therapeutic outcome-related ncRNAs by analyzing the differential gene expression of the exosomal RNA before and after treatment in individual patients. Consequently, the potential biological significance of exosomal RNA could be estimated by examining the associated mRNAs [18,19].

The employment of the aforementioned methodology prompted the postulation of a potential association between dysregulated inflammatory responses involving multiple microRNAs (miRNAs) and the incidence of early progression (EP) in patients with cervical cancer undergoing CCRT [19]. Nevertheless, the etiology underlying this phenomenon remained unidentified and is presumably attributed to the absence of the following considerations in prior research: first, although EP was initially established as the clinical endpoint, certain patients experiencing EP may survive for a prolonged duration and trigger a shift in the ultimate clinical endpoint to ED. Second, ncRNAs encompass a myriad of categories, including piwi-interacting RNA (piRNA), small nucleolar RNA (snoRNA), small nuclear RNA (snRNA), transfer RNA (tRNA), Y RNA (yRNA), long non-coding RNA (lncRNA), and more, as well as miRNA, which may contribute to mRNA regulation. Therefore, it is imperative to thoroughly evaluate every conceivable option. Third, considering that circulating cell-free miRNAs predominantly originate from blood cells, it is plausible to infer that exosomal ncRNAs are also potentially derived from blood cells [20]. Furthermore, prior clinical investigations have consistently demonstrated correlations between various factors, namely increased absolute neutrophil counts (ANC), elevated platelet counts (PLT), decreased hemoglobin (Hb), reduced absolute lymphocyte counts (ALC), heightened monocyte counts (Mo), and indicators such as increased neutrophil-to-lymphocyte ratio (NLR) = ANC/ALC, platelet-to-lymphocyte ratio (PLR) = PLT/ALC, and lymphocyte-to-monocyte ratio (LMR) = ALC/Mo; all of which are associated with adverse prognoses in solid tumors [21,22,23,24,25,26,27]. Consequently, a conjecture may be drawn regarding the existence of a potential link between these hematological parameters, exosome-derived ncRNAs, and ED.

Therefore, this study aimed to perform an integrative analysis of plasma exosomal ncRNAs in conjunction with blood cell dynamics to establish them as potential prognostic indicators of ED in CCRT-treated patients with cervical cancer.

## 2. Results

### 2.1. Elevated ANC Is Associated with ED in Patients with Cervical Cancer

A total of 370 cervical cancer patients were recruited for the following analysis after excluding the patients with RNA next-generation sequencing (NGS) data (*n* = 42, cohort 2) and those presenting with stage IB or IVB cervical cancer and having received CCRT at a total dose of less than 50 equivalent dose in 2 Gy fractions (*n* = 48) among the 460 patients treated with primary CCRT (Figure 1A). Among them, 70 patients who died and 112 patients who experienced disease progression were surveyed. Among the investigated subset, a cohort of 24 patients who experienced progression within 12 months of treatment and subsequent disease-specific death (DSD) within 15 months were assigned to the ED group (group 1), as shown in Figure 1B,C,E. Furthermore, 48 patients who underwent post-treatment completion progression and were survived beyond 30 months were assigned to the progression group (group 2), as shown in Figure 1B,C, and 102 patients who were followed up for more than 60 months without recurrence constituted the non-progression group (group 3), as depicted in Figure 1D. Following the implementation of propensity score matching (PSM) to the aforementioned 174 patients, 120 patients were included in cohort 1 (Figure 1A). Table 1 presents a comprehensive overview of the variations observed in complete blood counts (CBCs) among groups 1, 2, and 3 (Appendix A). Group 1 exhibited substantial correlations with increases in ANC, NLR, and PLR, whereas group 3 was associated with escalations in Hb and ALC. A comparison drawn between patients suffering from ED and those without ED (non-ED) revealed a correlation between elevated ANC and NLR values and ED in a subset of 42 patients constituting cohort 2 (Appendix A, Appendix A).

### 2.2. ncRNAs Associated with ED Were Selected from the CBC-Related Network 

Two networks were constructed using 55 ED- and CBC-associated RNAs, including 4 miRNAs, 2 piRNAs, 3 snoRNAs, 1 tRNA, 1 yRNA, 11 lncRNAs, and 33 mRNAs. The first and second networks displayed associations with Pearson correlation coefficients (R) of >0.5 and R > 0.6, respectively, between the RNAs (Figure 2A,B, respectively). Within these two networks, seven RNA components were identified as key elements; notably, three of them were ncRNAs, namely URS00001034C4 (piRNA), long intergenic non-protein coding (LINC)01003 (lncRNA), and miRNA (miR)-574-3p (Figure 2C,D).

### 2.3. RNAs Which Interplay with CBC Dynamics Are Associated with ED

Pretreatment CBC (CBC0) or $\sqrt{CBC0\times minimum\;CBC\;during\;CCRT\;(min\;CBC)}$ (CBC1) has a negative association with $\log\frac{CBCsecondweekduringCCRT(CBC2)}{CBC0}$ (CBC3), especially in ANC, Hb, and Mo (Appendix A). 

Among the RNAs associated with at least one of CBC0, CBC1, CBC2, CBC3, or min CBC (all CBC), Hb-related RNAs were the most common (48.3%), while ANC (26%), PLT (28.3%), and Mo (28.3%) had similar proportions, and ALC (21.9%) was the least common. Hb had the highest number of RNAs associated with CBC0, CBC1, CBC3, and min CBC, whereas ANC had the highest number of RNAs associated with CBC2 (Appendix A). The proportions of RNAs overlapping with at least two of the five CBC categories were 42%, 22%, 15%, 11%, 13%, and 20% for all CBC, CBC0, CBC1, CBC2, CBC3, and min CBC, respectively (Appendix A). 

Based on the number of RNAs associated with CBC0, CBC1, CBC2, and CBC3, the difference between the number of CBC0 and CBC2 (|CBC0-CBC2|) or the difference between the number of CBC1 and CBC2 (|CBC1-CBC2|) was positively correlated with the number of CBC3-associated RNAs (Appendix A). This implies that certain RNAs contribute to the negative association with either CBC0 and CBC3 or both CBC1 and CBC3 (CBC0(1) and CBC3). 

Based on this, I found that certain RNAs caused a negative correlation between CBC0(1) and CBC3. These included Hb (2087 cases), ANC (262 cases), Mo (201 cases), PLT (130 cases), and ALC (145 cases), as shown in Appendix A. Based on their association with ED, the ranking was as follows: Hb (60 cases), followed by ANC (10 cases), PLT (six cases), and Mo (five cases), as illustrated in Figure 3.

Given the rise in ANC or PLT among the ED patients as presented in Table 1, I organized RNAs that bring about changes in ANC or PLT, or changes in Hb or Mo, using solid and dashed lines, respectively. Furthermore, based on whether ED was associated with an increase or decrease in the logarithmic fold-change (log_2_FC) value of the RNA, they were demarcated using red and blue lines, resulting in a total of four types of RNAs.

### 2.4. RNAs Selected within the Network Can Potentially Predict ED in Patients with Cervical Cancer

Among the three selected ncRNAs from the two networks shown in Figure 2, miR-574-3p, which has a positive relationship with ED, was associated with ANC1 (CBC1 for ANC) and ANC3 (CBC3 for ANC) (Figure 3C), whereas LINC01003, which has a negative relationship with ED, was associated with Hb0 (CBC0 for Hb) and Hb3 (CBC3 for Hb) (Figure 3D). Among the 41 RNAs associated with CBC0, CBC1, CBC2, CBC3, or min CBC in the network, I identified acyl-coenzyme A thioesterase 9 (*ACOT9*) as the sole RNA uniquely associated with all five types of CBC, and its decrease was associated with ED (Figure 4). The miR-574-3p-LINC01003-*ACOT9* (log_2_FC) values obtained from the selected RNAs effectively distinguished the patients that experienced ED (Figure 5A and Appendix A). Figure 5B demonstrates that patients with miR-574-3p-LINC01003-*ACOT9* (log_2_FC) values less than 0 exhibited a substantially longer 30-month disease-specific survival (DSS) rate than patients with values greater than or equal to 0 (100% vs. 70.7%, *p* < 0.001). The median follow-up duration for cohort 2 was 34.5 months, and the 30-month DSS rate was 90.2%.

### 2.5. mRNAs Highly Relevant to Selected ncRNAs Are Linked to the Innate Immune System

Among the 7585 mRNAs that showed significant changes based on the log_2_FC of the selected RNAs, 219 common RNAs were associated with coagulation, as presented in Figure 6A. The top 100 mRNAs associated with miR-574-3p were deeply linked to neutrophil degranulation and the innate immune system pathway (Figure 6B). Similarly, the top 100 mRNAs related to LINC01003 were closely associated with the innate immune system pathway (Figure 6D). The mRNAs associated with *ACOT9* were deeply related to heme biosynthesis (Figure 6C). Notably, during the analysis of correlations among miR-574-3p-LINC01003-*ACOT9* (log_2_FC) and their constituent elements, the sequence emerged as *ACOT9* + LINC01003 (R^2^ = 0.9215), miR-574-3p−LINC01003 (R^2^ = 0.7257), and miR-574-3p–*ACOT9* (R^2^ = 0.2738), as presented in Appendix A. The biological ontology related to 47 RNAs in the network was the toll-like receptor (TLR)9 signaling pathway, and a key mRNA, interleukin 1 receptor-associated kinase 1 (*IRAK1*), located next to LINC01003 in the network, was closely associated with it (Figure 2A and Figure 7).

## 3. Discussion

The etiology underlying the premature disease progression-related death of patients despite timely cancer diagnosis and the implementation of stage-based treatment remains undetermined. The incidence of ED is not solely attributable to the stage and histological characteristics of cancer. It may occur when the host is incapable of executing their routine physiological functions. It was previously assumed that the issue revolved around the inflammatory response triggered by damage caused by cancer or its treatment. The current study was conducted on patients with cervical cancer who underwent primary CCRT. This study focused on exploring the causes by examining changes in CBC and plasma exosomal RNA in the peripheral blood of the host. Two selected ncRNAs reflecting ED were discovered to interact with the dynamics of neutrophils and Hb, and they were associated with the innate immune response.

To analyze the relationship between CBC changes and ED, I conservatively defined the ED group within cohort 1. CBCs between the ED and progression groups, the progression and non-progression groups, and between the ED and non-progression groups were compared after balancing the staging, RT field, and pathological findings. In this analysis, I confirmed that an increase in ANC, NLR, and PLR is associated with ED. When taking cohort 2 into account, it was observed that elevated neutrophil levels were more strongly linked to ED than increased PLT. The CBC1, defined as the geometric mean of CBC0 and min CBC, as well as min CBC, were employed to account for the effects of CBC on the treatment. It is worth noting that ANC0, min ANC, and ANC1 were all associated with ED, suggesting that ANC increased regardless of the treatment or cancer-related factors. The increase in ANC could be attributed to chronic inflammation due to sustained injuries, undetected infections, or other factors, as well as a decrease in the intrinsic ability to regulate ANC [28]. 

Other CBC parameters, such as Hb, ALC, and Mo, were associated with the non-progression or progression group, which could indirectly suggest a potential association with ED. Based on these results, I constructed a network using hematological factors such as ANC, PLT, Hb, ALC, Mo, NLR, PLR, and LMR, which have been previously associated with treatment outcomes for various cancers [21,22,23,24,25,26,27]. A comprehensive network using RNAs associated with both ED and these hematological parameters was created. To identify key factors among the RNA components within this network, three methods were employed. First, I calculated all possible combinations of log_2_FC values for RNA and their corresponding adjusted multiple correlation coefficients (adjusted R^2^). I then identified more than 50% of the ncRNAs included in the models and more than 60% of the mRNAs included in the models. Second, I selected RNAs (using log_2_FC) associated with the constancy of CBC as well as ED. The analysis revealed multiple RNAs with negative correlations between CBC0(1) and CBC3, with Hb, ANC, Mo, ALC, and PLT being the most abundant in that order. CBC3 was used to assess the degree of recovery after RT. Among these, the RNAs associated with ED accounted for approximately 2.9% of the total, with Hb-related RNAs making up 2.1%. Among the three ncRNAs selected from the network using the first method, miR-574-3p and LINC01003 were found to have negative correlations between ANC1 and ANC3 and between Hb0 and Hb3, which could be interpreted as a homeostatic mechanism for maintaining a quantitative balance within the blood. Third, I examined the number of specific CBC components that the RNAs in the network were associated with and identified *ACOT9* as a key factor. *ACOT9* was found to be associated with all ANC, PLT, Hb, Mo, and ALC. This suggests that *ACOT9* may be essential in the alterations in CBC parameters related to ED. 

The increase in miR-574-3p appears to be associated with scenarios in which ANC1 levels rise while ANC3 levels decline, which indicates its potential involvement in maintaining homeostasis in response to the increased post-RT ANC. It is plausible that miR-574-3p responds to a combination of factors, including dysregulation of ANC levels and external influences such as infection or tissue damage, in addition to the effects of RT, collectively contributing to its marked elevation. The decrease in LINC01003 may potentially be associated with scenarios in which Hb0 levels increase while Hb3 levels decrease, impacting the overall homeostasis of Hb. In situations where Hb declines owing to factors like treatment or cancer, maintaining or increasing LINC01003 may be a normal physiological response. Given that Hb0 and Hb3 are not directly linked to ED, the reduction in LINC01003 following RT may suggest a breakdown in homeostasis rather than being attributed to external factors. The decrease in *ACOT9* associated with ED is related to the decrease in Hb1, min Hb, Mo3, ALC1, ALC2, min ALC, min ANC, and min PLT. All these variables represent the impact exerted by CBC on treatment. This suggests that *ACOT9* may possess a function related to the production of CBC components. 

To estimate the potential functions of the selected RNAs, I identified the mRNAs that were significantly associated with the log_2_FC of the three RNAs. I then conducted an enrichment analysis using mRNAs that were commonly associated across all three RNAs, as well as the mRNAs that were specifically associated with each individual RNA. The mRNAs commonly associated with all three RNAs were indeed linked to platelet activation, a critical component of the innate immune response [29,30]. mRNAs associated with miR-574-3p have been linked to neutrophil degranulation induced by TLR signaling, a component of the innate immune response with implications for both pathogen destruction and potential contributions to cancer metastasis [31,32,33]. In the network, the increased expression of miR-574-3p showed a strong correlation with the upregulation of the cluster of differentiation 274 (*CD274*), commonly known as programmed death-ligand 1 (PD-L1). This could potentially help stabilize the elevated neutrophil levels by reducing the elevated ANC, as indicated in a report that highlights how PD-L1 on elevated neutrophils inhibits their migration within the bone marrow, possibly due to its anti-fungal properties [34]. Considering the connection between the increase in ANC associated with ED and the increase in miR-574-3p, it is plausible that elevated miR-574-3p may be associated with the proximity to a chronic inflammatory environment, such as sustained infection, high levels of stress, unhealthy eating habits, smoking, or excessive alcohol consumption, to facilitate cancer growth [35]. In fact, neutrophil miR-574-3p has been reported as a diagnostic marker for lung cancer [36]. 

LINC01003 is a gene associated with the regulation of Hb levels, and it might have a role in the innate immune response. Although there have not been any specific reports linking LINC01003 to CBC or the innate immune response, some inferences can be drawn from existing research. One notable aspect is its proximity to *IRAK1*, a gene involved in the TLR9 signaling pathway, which suggests that LINC01003 could be connected to the immune surveillance function of red blood cells (RBCs) [33,37]. A TLR9 was found to be expressed on the surface of RBCs, denoted as RBC-TLR9, which has the capability to bind to specific regions of mitochondrial DNA (mtDNA) known as CpG regions [38]. This interaction has the potential to trigger an innate immune response, particularly related to conditions such as infections, cancer, trauma, and cardiovascular diseases [38,39]. For instance, in critical illnesses like sepsis, increased binding of mtDNA to RBC-TLR9 can lead to anemia due to the clearance of RBCs [38]. However, under normal circumstances, the levels of RBC-TLR9 may be typically well-regulated. Notably, many lncRNAs, including lincRNA-cyclooxygenase (Cox) and myocardial infraction associated transcript 2 (Mirt2), have been shown to influence the expression of immune-related genes following TLR stimulation in macrophages [33]. Therefore, reduced levels of LINC01003 might play a crucial role in the dysregulated inflammation that could occur in response to TLR9 stimulation, particularly in situations where RBC-TLR9 levels are reduced due to treatments or cancer. Consequently, tumors may progress rapidly, even when receiving appropriate treatment, if changes in cancer mtDNA, which is closely linked to the plasticity of cancer cells, cannot be detected due to insufficient levels of RBC-TLR9 following a reduction in LINC01003 [40,41]. 

Concerning the quantitative aspect of RBCs, several lncRNAs have been reported to regulate RBC development [42]. Additionally, RBCs act as scavengers of chemokines to curtail excessive inflammation and exhibit antibacterial effects through Hb [39]. Furthermore, RBC-derived exosomes have been demonstrated to initiate proinflammatory processes by transferring to Mo and directly contributing to antigen presentation for the adaptive immune response [43,44]. Therefore, the reduction in LINC01003 following RT may be linked to the inadequate recovery of diminished RBC levels post-RT, potentially hindering the activation of innate immune responses against cancer cells and thereby facilitating cancer progression. In the present study, LINC01003 displayed the strongest correlation with ED among the three RNAs in the miR-574-3p-LINC01003-*ACOT9* (log_2_FC).

Considering that *ACOT9* is associated with all CBC components, the production of fatty acids from *ACOT9* may potentially serve as a crucial energy source for hematopoiesis [45,46]. Therefore, it can be assumed that exosomal *ACOT9* is delivered to the bone marrow when CBC is insufficient during CCRT. Owing to the high proportion of RNA associated with Hb, there may be a notable correlation between the RNAs associated with *ACOT9* and heme synthesis. Furthermore, the fact that the combination of three RNAs was best explained when *ACOT9* was linked to LINC01003, rather than miR-574-3p, supports this analysis. 

Through an integrated analysis of changes in plasma exosomal RNA and CBC, the hypothesis that the innate immune response of RBCs regulated by lncRNAs is associated with ED was suggested as a fresh perspective on the causes of ED. Methodologically, using RNA log_2_FC values obtained by comparing RNA read counts before and after CCRT for each patient enables a comprehensive analysis of RNAs and CBCs, facilitating the assessment of their impact. LIN01003, associated with Hb (the largest subset of CBC-related RNAs) homeostasis, exhibited the strongest association with ED among the three selected RNAs. Additionally, LINC01003 combined with *ACOT9*, linked to heme biosynthesis, provided the most comprehensive explanation for miR-574-3p-LINC01003-*ACOT9* (log_2_FC). These findings highlight the importance of investigating the role of extracellular vesicular lncRNAs originating from RBCs to understand the underlying causes of ED. 

Nonetheless, this study has some limitations. First, it is essential to conduct in vivo and in vitro biological validation of the assumptions derived from this research, particularly in elucidating the roles of the RNAs contained in the EVs originating from neutrophils and RBCs. This would establish the biological basis for the ncRNAs presented in this study. Second, the timing of ED may vary depending on the cancer type (such as breast, lung, or head and neck cancer) and the specific treatments (surgery, chemotherapy, or RT) administered as primary or subsequent interventions. Therefore, the definition of “early death” should be discussed and refined by expert groups within each cancer category. Finally, to uncover the underlying causes of ED and establish a solid foundation for cancer treatment, seamless collaboration between researchers and clinicians is paramount.

In conclusion, this study investigates the perplexing issue of ED in patients with cancer, looking beyond traditional factors such as cancer stage and pathology. Focusing on patients with cervical cancer undergoing primary CCRT, I discovered two key ncRNAs, miR-574-3p and LINC01003, and one mRNA, *ACOT9*, associated with innate immune response modulation. However, further biological validation, customized ED definitions for different cancer types and treatments, and interdisciplinary collaboration should be warranted to comprehensively enunciate the complexities associated with ED.

## 4. Materials and Methods

### 4.1. Study Participants 

A cohort of 460 patients diagnosed with 2018 International Federation of Gynecology and Obstetrics stage I-IVB cervical cancer received weekly cisplatin-based CCRT at the Department of Radiation Oncology, Ajou University Hospital, between April 2001 and April 2023. Among them, blood samples were collected from a subset of 42 patients prior to and two weeks after RT to acquire plasma exosomal RNA NGS data. A matching analysis was conducted on 370 patients after excluding the patients with RNA NGS data (*n* = 42) and those presenting with stage IB or IVB cervical cancer and having received CCRT at a total dose of less than 50 equivalent dose in 2 Gy fractions (Figure 1). Detailed information regarding the treatment procedure and variables is documented in the Appendix A.

### 4.2. Blood Cell Dynamics (CBC) 

A CBC includes ANC, PLT, Hb, ALC, Mo, NLR, PLR, and LMR. CBC0 refers to the CBC recorded prior to treatment and is represented as ANC0, PLT0, Hb0, ALC0, Mo0, NLR0 ($\frac{ANC0}{ALC0}$), PLR0 ($\frac{PLT0}{ALC0}$), and NLR0 ($\frac{ALC0}{Mo0}$). Min CBC signifies the minimum CBC values during CCRT. Min ANC, min PLT, min Hb, min ALC, and min Mo are used. CBC1 is depicted by the geometric mean of CBC0 and min CBC ($\sqrt{CBC0\times min\;CBC})$ and is represented as ANC1, PLT1, Hb1, ALC1, Mo1, NLR1 ($\frac{ANC1}{ALC1}$), PLR1 ($\frac{PLT1}{ALC1}$), and LMR1 ($\frac{ALC1}{Mo1}$). CBC2 indicates the CBC recorded during the second week of CCRT and is denoted as ANC2, PLT2, Hb2, ALC2, and Mo2. CBC3 is defined as the logarithmic change of CBC2 relative to CBC0 ($\log\frac{CBC2}{CBC0}$), and is represented as ANC3, PLT3, Hb3, ALC3, and Mo3.

### 4.3. Comparison of CBCs between Groups Based on the Timing of DSD

A comparative analysis was performed on a subgroup of patients experiencing progression or DSD. This analysis encompassed the temporal relationship between DSD and disease progression, the frequency of DSD occurrences, the incidence rates of disease progression, and a comparison of the DSD timing between patients who experienced progression within one year and those who did not. Patients were selected from the ED (group 1), progression (group 2), and non-progression (group 3) groups based on the aforementioned parameters. Subsequently, a 1:1 PSM was conducted between groups 1 and 2, groups 2 and 3, as well as groups 1 and 3, with a focus on the pathology, stage, and RT of the cervical cancer. The variations in CBC among these matched groups were compared, and the patients included in this comparison were classified as cohort 1. For cohort 2 patients, a 1:2 PSM was performed based on age, pathology, and stage between ED and non-ED patients, and the differences in CBC between these two groups were compared. For categorical data, a Chi-Square test was used, while for continuous data, a normality test was initially conducted, followed by either a *t*-test or a Wilcoxon test.

### 4.4. Log_2_FC of RNAs 

The Log_2_FC of the plasma exosomal RNAs was examined using NGS data obtained from plasma exosomes, encompassing information on ncRNAs and mRNAs. RNAs that remained undetected in at least 50% of the samples were excluded, and a total of 508 miRNAs, 352 piRNAs, 629 snoRNAs, 2000 snRNAs, 2523 tRNAs, 700 yRNAs, 3151 lncRNAs, and 14,908 mRNAs were subjected to analysis. The Log_2_FC values were calculated by comparing the read counts of RNAs prior to (control) and after the second week of CCRT (treatment), following a trimmed mean of M-value normalization using edgeR for the 42 cohort 2 patients. Plasma exosomal RNA sequencing and profiling were performed using Macrogen and ROKIT Genomics (www.macrogen.com, accessed on 1 October 2018, and www.rokitgenomics.com, accessed on 1 March 2021, respectively). Further details are delineated in the Appendix A.

### 4.5. Network Construction

I defined a link with ANC, min ANC, or ANC1 if it was associated with one or more of them, and similarly applied this to PTL, Hb, ALC, and Mo. For NLR, I defined a link with NLR if it was associated with NLR0 or NLR1 and extended this to PLR and LMR. Based on these definitions, I constructed the network using exosomal RNA values (log_2_FC) that satisfied either of the following two conditions: (1) exosomal RNA values associated significantly with three or more of ANC, PLT, Hb, ALC, Mo, NLR, PLR, and LMR and significantly linked to both ED (|correlation coefficient (R)| > 0) and EP (|R| > 0), or (2) exosomal RNA values strongly associated with ED (|R| > 0.5). Network analysis and visualization were conducted using the igraph package.

### 4.6. Association between CBC Alterations and RNA Expression

The correlation between CBC0 (except NLR0, PLR0, and LMR0), CBC1 (except NLR1 PLR1, and LMR1), CBC2, CBC3, and min CBC was analyzed using correlation matrixes for cohorts 1 and 2, respectively. After identifying the RNAs that were significantly associated with CBC0, CBC1, CBC2, CBC3, or min CBC in cohort 2, Venn diagrams were constructed based on ANC, PLT, Hb, ALC, and Mo. From these diagrams, the number of RNAs associated with CBC0, CBC1, CBC2, and CBC3, as well as the number associated with min CBC, were determined. Additionally, a correlation analysis was performed between |CBC0-CBC2| or |CBC1-CBC2| and CBC3. For all the RNAs included in the analysis, the number of RNAs associated with either CBC0 and CBC3 or both CBC1 and CBC3 (CBC0(1) and CBC3) was described for each type of CBC. As a secondary step, the RNAs associated with ED were extracted from these sets. The matrix of Pearson’s correlations among all CBCs, all RNAs, and ED was calculated using the recorr function in the Hmisc package.

### 4.7. Selection of RNAs as Prognostic Biomarkers of ED

On the network, the RNAs associated with at least one of ANC0, ANC1, ANC2, ANC3, or min ANC were defined as being associated with ANC. The same definition was extended to PLT, Hb, ALC, and Mo. Venn diagrams were generated for ANC, PLT, Hb, ALC, and Mo based on the RNAs associated with the CBC. All types of CBC-associated RNA were selected. The key components among the RNAs in the network were identified through an exhaustive search using the regsubsets function in the leaps package for R. From these RNAs, the ncRNAs associated with CBC0(1) and CBC3 were selected.

### 4.8. Survival Analysis 

The Wilcoxon rank-sum test was utilized to compare the sum and difference of the log_2_ FC values of the selected RNAs between the ED and non-ED patients, which subsequently served as the basis for the collation of the two subsets using Kaplan–Meier plots and the log-rank test.

### 4.9. Biological Function of Selected RNAs

An enrichment analysis and visualization were performed for pathway and gene ontology annotation in Enrichr (https://maayanlab.cloud/Enrichr accessed on 1 May 2023) using the mRNAs that significantly changed based on the log_2_FC of the selected RNAs, as well as the RNAs from the network [47]. 

All data analysis and visualization were conducted using R version 4.2.3 (https://www.r-project.org, accessed on 1 December 2022).

## Figures and Tables

**Figure 1 ijms-25-00126-f001:**
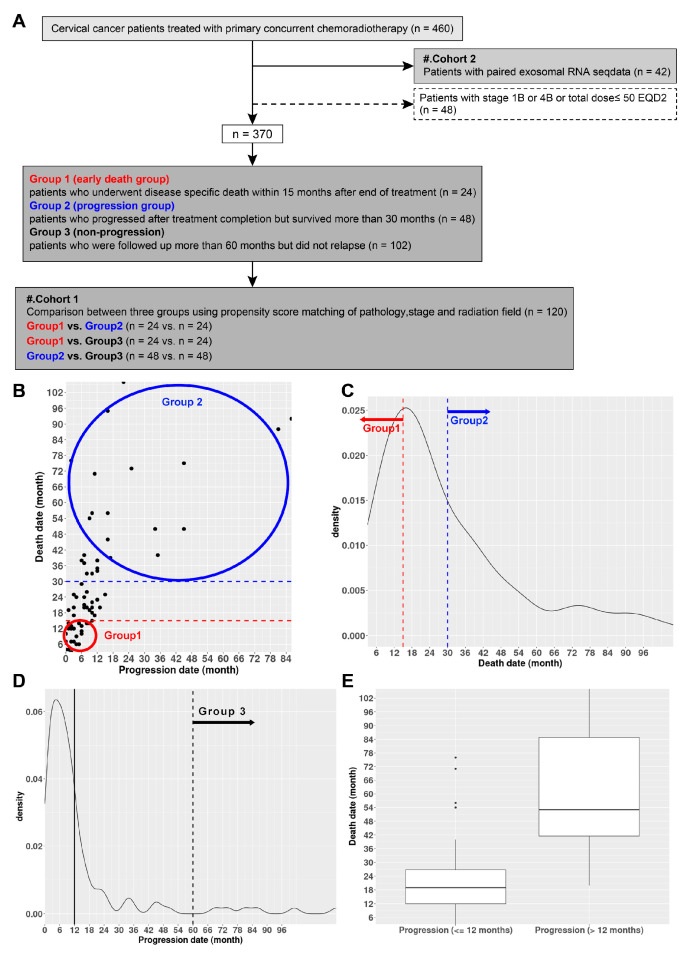
(**A**) Classification diagram of clinical data used for this study. (**B**) Scatter plot of disease progression versus death time (in months). (**C**) Density plot illustrating the death times of 70 patients who died among the initially recruited 370 patients. (**D**) Density plot representing the post-treatment disease progression times of 112 patients. (**E**) Boxplot comparing the survival times of patients who experienced disease progression within 12 months of treatment completion and those who experienced it thereafter. EQD2: equivalent dose in 2 Gy fractions.

**Figure 2 ijms-25-00126-f002:**
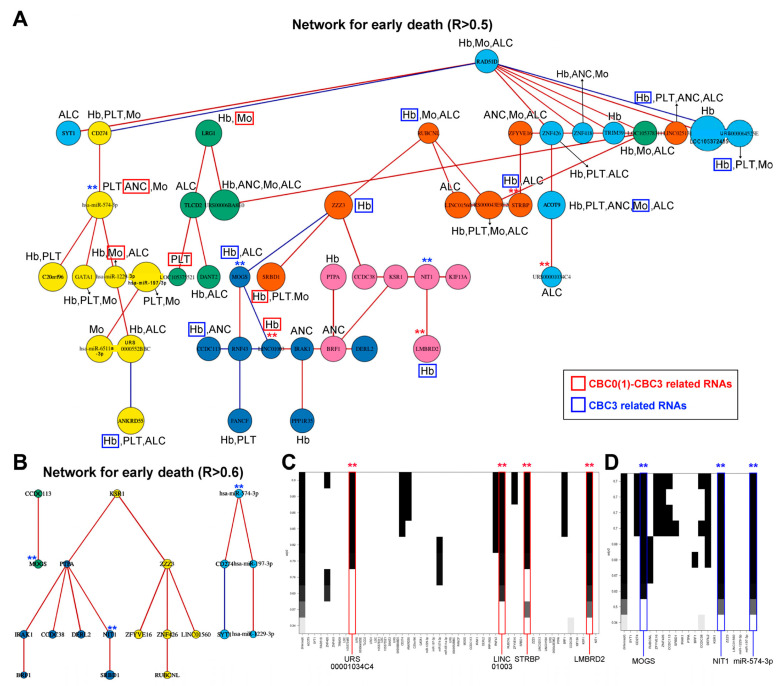
Networks constructed from early death-related RNAs associated with complete blood counts (CBC) and selection of statistically representative RNAs. (**A**) A network comprising 47 RNAs with a correlation coefficient of R > 0.5, and (**B**) network of 20 RNAs with a correlation coefficient of R > 0.6, relevant to early death and hematological parameters. (**C**) One lncRNA, one piRNA, and two mRNAs, selected after obtaining adjusted R^2^ values from multiple linear regressions of all possible combinations of RNAs included in the network (R > 0.5). (**D**) One miRNA and two mRNAs, selected after obtaining adjusted R values from multiple linear regressions of all possible combinations of RNAs included in the network (R > 0.6). ANC: absolute neutrophil counts, PLT: platelet, Hb: hemoglobin, ALC: absolute lymphocyte counts, Mo: monocyte, ncRNA: long non-coding RNA, piRNA: piwi-interacting RNA, mRNA: messenger RNA, miRNA: microRNA, CBCs (ANC, PLT, Hb, ALC, Mo) above RNAs on network associated with pretreatment CBC (CBC0), minimum CBC during chemo-radiotherapy (min CBC), $\sqrt{CBC0\times min\;CBC}$ (CBC1), CBC second week during chemo-radiotherapy (CBC2), or $\log\frac{CBC2}{CBC0}$ (CBC3) are presented. CBCs with red boxes indicate RNAs correlated with either CBC0 and CBC3 or both CBC1 and CBC3 (CBC0(1) and CBC3), whereas those with blue boxes indicate RNAs correlated with CBC3 alone. Red asterisks (**) above RNAs selected from network (R > 0.5) and blue asterisks (**) above RNAs selected from network (R > 0.6) are presented.

**Figure 3 ijms-25-00126-f003:**
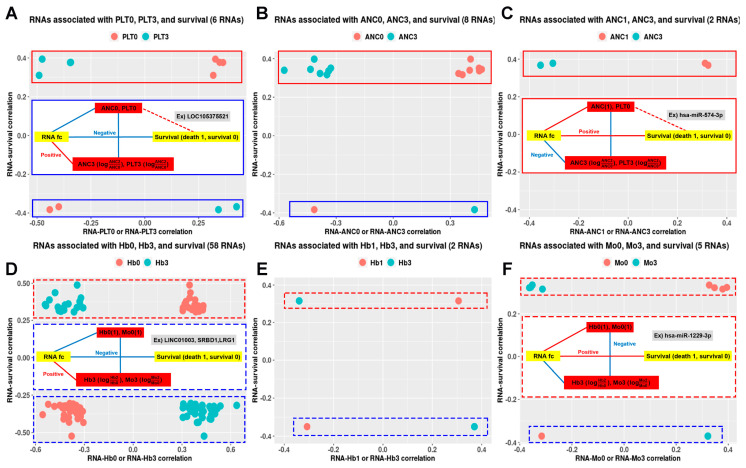
Scatterplots between early death (ED)-related exosomal RNAs relevant to either CBC0 and CBC3 or both CBC1 and CBC3 (CBC0(1) and CBC3) selected under the condition that CBC0(1) and CBC3 were negatively correlated. Scatterplots (**A**) between RNA−PLT0 or RNA−PLT3 and RNA−survival correlation coefficients, (**B**) RNA−ANC0 or RNA−ANC3 and RNA–survival correlation coefficients, (**C**) RNA−ANC1 or RNA−ANC3 and RNA–survival correlation coefficients, (**D**) RNA−Hb0 or RNA−Hb3 and RNA–survival correlation coefficients, (**E**) RNA−Hb1 or RNA−Hb3 and RNA–survival correlation coefficients, and (**F**) RNA−Mo0 or RNA−Mo3 and RNA−survival correlation coefficients. In the center of (**A**), the solid blue line outlines a diagram with LOC105375521 as a PLT0-related RNA, indicating a negative association with ED, particularly concerning increased PLT. Similarly, in (**C**), the solid red line surrounds a diagram featuring miR-574-3p as an ANC1-related RNA, showing a positive association with ED, especially related to increased ANC. In the center of (**D**), the dashed blue line outlines a diagram with LINC01003, S1 RNA binding domain 1 (*SRBD1*), and leucine-rich α-2 glycoprotein 1 (*LRG1*) as Hb0-related RNA, showing a negative association with ED. Similarly, in (**F**), the dashed red line surrounds a diagram featuring miR-1229-3p as Mo0-related RNA, indicating a positive association with ED. PLT: platelet, ANC: absolute neutrophil counts, Hb: hemoglobin, Mo: monocyte, CBC: complete blood counts, CBC0: pretreatment CBC, min CBC: minimum CBC during chemo-radiotherapy, CBC1: $\sqrt{CBC0\times min\;CBC}$, CBC2: CBC second week during chemo-radiotherapy. CBC3: $\log\frac{CBC2}{CBC0}$. Red rectangles indicate RNAs positively correlated with ED, whereas blue rectangles represent RNAs negatively correlated with ED. Rectangular solid lines indicate RNAs associated with ANC or PLT, whereas Hb- or Mo-related RNAs are presented as rectangular dashed lines.

**Figure 4 ijms-25-00126-f004:**
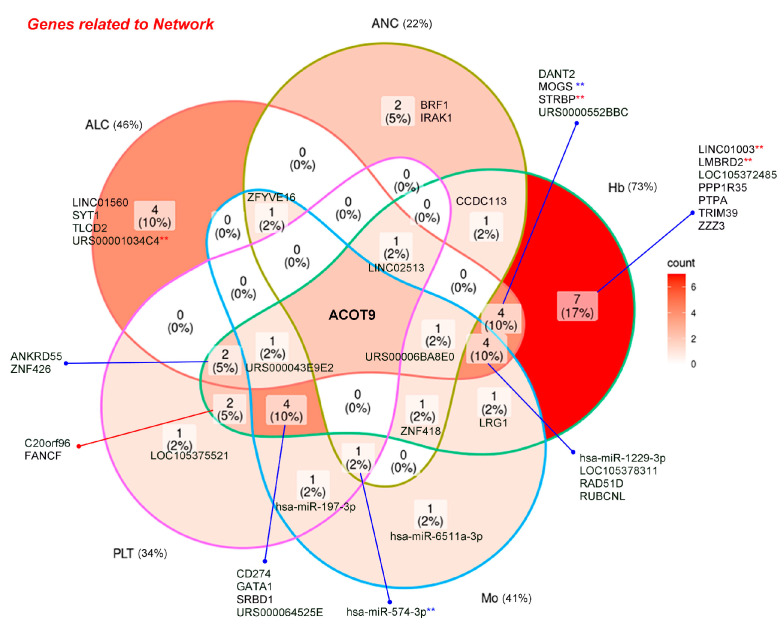
Venn diagram of 41 RNAs in the network (R > 0.5) according to the association among ANC (absolute neutrophil counts), PLT (platelet), Hb (hemoglobin), ALC (absolute lymphocyte count), and Mo (monocyte). Red asterisks (**) above RNAs selected from network (R > 0.5) and blue asterisks (**) above RNAs selected from network (R > 0.6) are presented.

**Figure 5 ijms-25-00126-f005:**
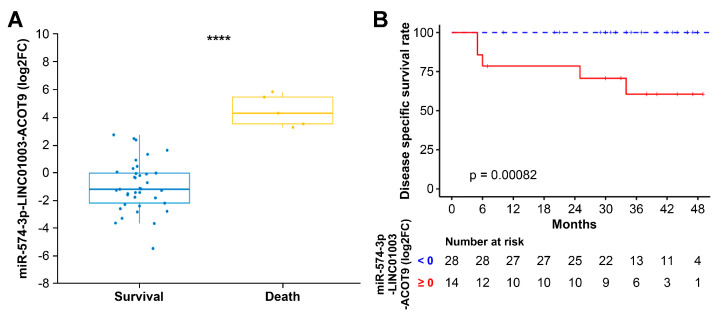
Comparison of survival between the two groups according to zero value of the miR-574-3p-LINC01003-*ACOT9* (log_2_FC). (**A**) The miR-574-3p-LINC01003-*ACOT9* (log_2_FC) exhibited significant differences between the two groups based on whether the patients experienced cervical cancer-related deaths. (**B**) Kaplan–Meier plots and log-rank tests were performed for comparing the disease-specific survival of patients with miR-574-3p-LINC01003-*ACOT9* < 0 and patients with miR-574-3p-LINC01003-*ACOT9* ≥ 0. Log_2_ fold change: log_2_FC, **** *p* < 0.0001.

**Figure 6 ijms-25-00126-f006:**
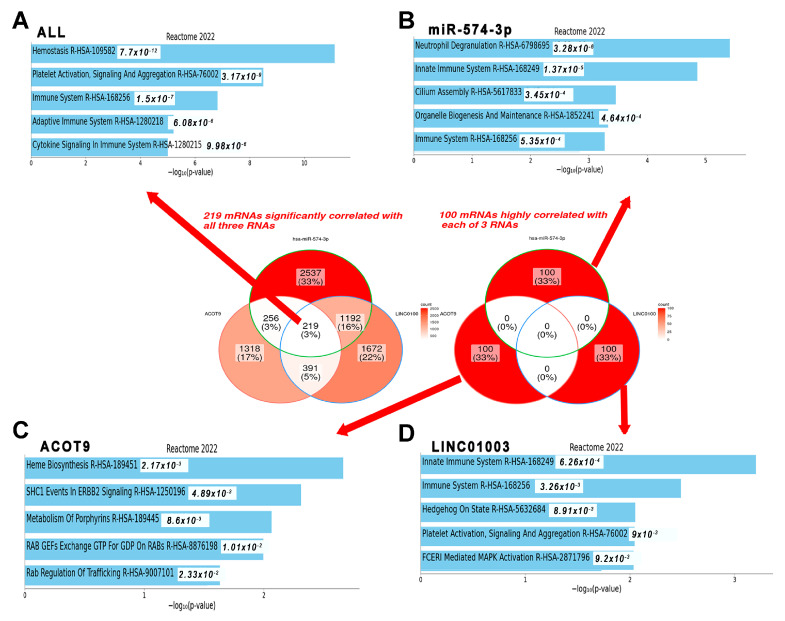
Pathway enrichment analysis using mRNAs related to miR-574-3p, *ACOT9*, and LINC01003. Pathway annotations were documented for (**A**) the overlap of mRNAs associated with the three RNAs, as well as the mRNAs highly correlated with (**B**) miR-574-3p, (**C**) *ACOT9*, and (**D**) LINC01003. mRNA: messenger RNA.

**Figure 7 ijms-25-00126-f007:**
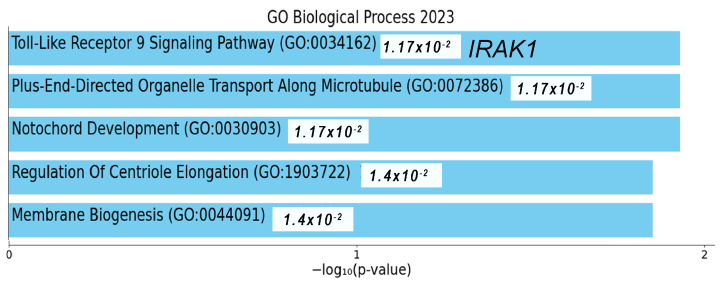
Ontology enrichment analysis using 47 RNAs on the constructed network (R > 0.5).

**Table 1 ijms-25-00126-t001:** Summary of comparison results of hematological parameters between group 1 (early death), group 2 (progression), and group 3 (non-progression) through propensity score matching (PSM).

PSM	Time	ANC	PLT	Hb	ALC	Mo	NLR	PLR	LMR
Group 1	CBC0	●(↑) (↓)					●(↑) (↓)	●(↑) (↓)	
versus	Min CBC	●(↑) (↓)							
Group 2	CBC1	●(↑) (↓)					●(↑) (↓)		
Group 1	CBC0			▲(↓) (↑)	●(↓) (↑)		●(↑) (↓)	●(↑) (↓)	
versus	Min CBC	●(↑) (↓)	●(↑) (↓)						
Group 3	CBC1	●(↑) (↓)	●(↑) (↓)				●(↑) (↓)	●(↑) (↓)	
Group 2	CBC0			●(↓) (↑)	▲(↓) (↑)	●(↓) (↑)			
versus	Min CBC			▲(↓) (↑)		▲(↓) (↑)			
Group 3	CBC1			●(↓) (↑)	●(↓) (↑)	●(↓) (↑)			●(↑) (↓)

PSM: propensity score matching; ANC: absolute neutrophil count (cells/μL); PLT: platelet $\times 10^{-3}$ (cells/μL); Hb: hemoglobin (g/dL); ALC: absolute lymphocyte count (cells/μL); Mo: monocyte (cells/μL); NLR: ANC/ALC; PLR: PLT/ALC; LMR: ALC/Mo; CBC: complete bleed count (ANC, PTL, Hb, ALC, or Mo); CBC0: pretreatment CBC (ANC0, PLT0, etc.); Min CBC: minimum CBC during chemo-radiotherapy (min ANC, min PLT, etc.); CBC1:$\sqrt{CBC0\times min\;CBC}$ (ANC1, PLT1, etc.); CBC0 for NLR, PLR; LMR: ANC0/ALC0, PLT0/ALC0, and ALC0/Mo0; CBC1 for NLR, PLR; LMR: ANC1/ANC1, PLT1/ ALC1, and ALC1/ Mo1; ● *p* < 0.05; ▲ *p* < 0.1. Red, blue, and black letters and arrows represented group 1, group 2, and group 3, respectively.

## Data Availability

The clinical and processed RNA data documented in this study are available at: https://github.com/oyeoncho/ed, accessed on 1 September 2023. All data pertaining to this study are available as raw sequencing data at: ArrayExpress (accession numbers: E-MTAB-10215, 10930, 12187). All R codes used for this analysis are available at: https://github.com/oyeoncho/ed, accessed on 1 September 2023.

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
