# Peer review of "Post-Radiotherapy Exosomal Non-Coding RNA and Hemograms for Early Death Prediction in Patients with Cervical Cancer"

_ijms, 2023, doi:10.3390/ijms25010126_

Round 1

Reviewer 1 Report

Comments and Suggestions for Authors

The study aimed to identify markers associated with the early death in patients undergoing chemo-radiotherapy. These markers were non-coding RNAs from exosomes associated with blood cells, which could have prognostic significance in cervical cancer. The work is interesting and could have a significant impact. However, in my opinion, major revisions and validation of the biomarkers are required. In the abstract, the features of the considered cohorts of patients are not properly described. In general, the description of the study should also be improved, for example there is no mention of the methodology used.

1.       The introduction should better reflect the current state of the art on the topic. No references from the literature regarding ncRNAs are described. A paragraph could be included to better introduce and precede the paragraph on exosomes.

2.       The figures are of very poor quality, with some being illegible. The reader (including myself) is unable to verify the claims made. These figures cannot be accepted in their current form.

3.       The description of the groups in the 2.1 paragraph of the Results is not fluid and clear. I suggest including Supplementary Figure 1 in the manuscript after reworking it with Figure 1 for better understanding. Furthermore, it is unclear what the 460 samples refer to.

4.       Supplement methods report that the analysis was carried out on 10 ng of RNA. This amount seems very small. Can the author provide a reference indicating that this amount is adequate?

5.       For transparency, the author should include the entire list of deregulated genes (mRNAs and ncRNAs) that emerge from the analysis and comparison between the paired groups, indicating the fold, p value, FDR value etc... in a supplementary file.

6.       A critical point concerns the exclusion of RNAs that remained undetected in at least 50% of the samples. This could lead to the loss of relevant data, if the deleted RNAs are attributed to a specific cohort. The author must verify if relevant data has not been excluded. If the value of 1 read is assigned to the absent RNAs, do other possible markers emerge?

7.       The 2.3 paragraph of Results is heavy to read. It would be very important to make it more attractive and better describe figure 3 and the relevance of the data.

8.       Finally, Finally, it is not clear from the manuscript whether the biomarkers (miR-574-3p-LINC01003-ACOT9) have been validated subsequently, such as using another analytical method like RT-PCR, on new patients in association with their hemograms. This validation is necessary. If it was done, it is not mentioned in the manuscript.

Minor revision

There are several punctuation errors.

Line 102, the legend below Table 1 is not centered.

The format of the references must be the same for all and include the last page

Comments on the Quality of English Language

Some paragraphs can be written more fluently.

Author Response

The study aimed to identify markers associated with the early death in patients undergoing chemo-radiotherapy. These markers were non-coding RNAs from exosomes associated with blood cells, which could have prognostic significance in cervical cancer. The work is interesting and could have a significant impact. However, in my opinion, major revisions and validation of the biomarkers are required. In the abstract, the features of the considered cohorts of patients are not properly described. In general, the description of the study should also be improved, for example there is no mention of the methodology used.

I have revised the abstract per your suggestions (page 1).

  1. The introduction should better reflect the current state of the art on the topic. No references from the literature regarding ncRNAs are described. A paragraph could be included to better introduce and precede the paragraph on exosomes.

I have added a paragraph related to ncRNA relevant for both inflammation and cancer (page 2).

  1. The figures are of very poor quality, with some being illegible. The reader (including myself) is unable to verify the claims made. These figures cannot be accepted in their current form.

I have replaced all figures with those having better quality.

  1. The description of the groups in the 2.1 paragraph of the Results is not fluid and clear. I suggest including Supplementary Figure 1 in the manuscript after reworking it with Figure 1 for better understanding. Furthermore, it is unclear what the 460 samples refer to.

I have revised Figure 1 per your suggestion. The relevant explanation including description of 460 patients has been added in section 2.1 of Results. (page 2-3).

  1. Supplement methods report that the analysis was carried out on 10 ng of RNA. This amount seems very small. Can the author provide a reference indicating that this amount is adequate?

Thank you for this comment. We searched the literature and found that all the following studies used 10 ng RNA and performed small RNAseq in almost the same way.

Kyue-Yim et al., Molecular Signature of Extracellular Vesicular Small Non-Coding RNAs Derived from Cerebrospinal Fluid of Leptomeningeal Metastasis Patients: Functional Implication of miR-21 and Other Small RNAs in Cancer Malignancy. Cancers 2021, 13, 209. https://doi.org/10.3390/cancers13020209

Dughyun Choi et al., Weight Change Alters the Small RNA Profile of Urinary Extracellular Vesicles in Obesity. Obes Facts 22 March 2022; 15 (2): 292–301. https://doi.org/10.1159/000521730

Youn Jae Jung et al., Cell reprogramming using extracellular vesicles from differentiating stem cells into white/beige adipocytes.Sci. Adv.6,eaay6721(2020).DOI:10.1126/sciadv.aay6721

Additionally, it is mentioned on the website of Takara (https://www.takarabio.com/products/next-generation-sequencing/epigenetics-and-small-rna-sequencing/small-rna-seq-kit), which sells cDNA synthesis kits, that data can be generated with 1 ng to 2 ug of total RNA input.

  1. For transparency, the author should include the entire list of deregulated genes (mRNAs and ncRNAs) that emerge from the analysis and comparison between the paired groups, indicating the fold, p value, FDR value etc... in a supplementary file.

As the supplementary file was too large, these data were not submitted in the supplementary file but were saved as rt_fc2.Rdata in the process_data folder (https://github.com/oyeoncho/ed). All the above-mentioned values can be accessed from this folder. We apologize for the inconvenience caused due to this.

  1. A critical point concerns the exclusion of RNAs that remained undetected in at least 50% of the samples. This could lead to the loss of relevant data, if the deleted RNAs are attributed to a specific cohort. The author must verify if relevant data has not been excluded. If the value of 1 read is assigned to the absent RNAs, do other possible markers emerge?

Thank you for this comment. Data sparsity due to many zero read counts (undetected read counts) in the comparison group can distort the results of differential gene expression; this is a major problem associated with single cell RNA-seq data wherein zero inflation is prominent (Rouchen Jian et al., Statistics or biology: the zero-inflation controversy about scRNA-seq data. Genome Biol 23, 31 (2022). https://doi.org/10.1186/s13059-022-02601-5).

Removing RNAs with a high number of zero read counts may result in loss of biological information; however, this method facilitated statistically appropriate comparisons in this study using log2 FC values. In this study, various types of RNA NGS data were analyzed. The zero read count percentage among various RNA types varied in the range of 11.8–98.6% (standard deviation (SD): 30.4%), so the log2FC values ​​analyzed from this were not calculated under equivalent conditions for each RNA type; however, if RNAs remain undetected in at least 50% of the samples, 50% of the samples are excluded and the range can be reduced to 4.7–28.5% (SD: 6.9%), greatly reducing the difference of zero read count percentage among various RNA types. Therefore, the calculation of log2FC using the 50% filtered dataset can be more meaningful in this study.

Filter

Zero read count percentage

miRNA

piRNA

snoRNA

snRNA

tRNA

yRNA

mRNA

lncRNA

SD

0%

77.4

98.6

66.1

35

11.8

12.5

26.1

69.1

30.4

50%

16.6

15

21.9

19.6

4.7

9.9

12.9

28.5

6.9

        For all types of RNA, when log2FC values ​​were obtained for a dataset with 100% of undetected reads removed, 50% of the data were removed. The more the undetected read counts are removed, the more bell-shaped the density function for log2FC values ​​becomes. It shows the characteristics of a normal distribution, and the diversity shows a decreasing trend. If the undetected read counts are not removed, one may include biological meaning, but it deviates from statistical assumptions about the population. Therefore, the 50% filtered dataset can appropriately satisfy the statistical distribution and biological significance of RNAs.

        The figure below shows density plots of log2FC values ​​of miRNA, lncRNA, and mRNA. The red circled areas clearly show normal distribution and diversity of log2FC values.

Before conducting the current study, I preliminarily checked whether there was a difference between the density functions for zero read counts according to clinical variables such as stage, pathology, disease progression, acute response, and complete response (cr) for all RNA types. There was no difference or change in the zero read count density of the groups according to clinical variables around the average, rather than in areas with a large number of zero read counts. Therefore, since the filtered RNAs are likely to be associated with the changes in areas with a large number of zero read counts, it can be inferred that there is a small possibility that clinically meaningful RNA will be excluded through filtering of the zero read counts.

The following are the density plots of lncRNA with zero read counts according to various clinical variables.

  1. The 2.3 paragraph of Results is heavy to read. It would be very important to make it more attractive and better describe figure 3 and the relevance of the data.

I have revised section 2.3 and figure 3 legend for clarity and readability. (page 5-6)

  1. Finally, it is not clear from the manuscript whether the biomarkers (miR-574-3p-LINC01003-ACOT9) have been validated subsequently, such as using another analytical method like RT-PCR, on new patients in association with their hemograms. This validation is necessary. If it was done, it is not mentioned in the manuscript.

 We agree that it is necessary to validate the DE genes via RT-qPCR analysis. We plan to test the validation set after collecting samples from the Gynecological Cancer Center at our institution so that we have necessary number and amount of plasma samples from cervical cancer patients in a future study.

Minor revision

There are several punctuation errors.

To ensure no such errors exist, I have got the manuscript proofread by a professional English editing company.

Line 102, the legend below Table 1 is not cantered.

I revised this.

The format of the references must be the same for all and include the last page

I revised the reference format.

Reviewer 2 Report

Comments and Suggestions for Authors

In the present study, the author performed an integrative analysis of plasma exosomal ncRNAs in conjunction with blood cell dynamics to know their potential prognostic indicators for early death (ED) in concurrent chemo-radiotherapy (CCRT)-treated patients with cervical cancer. The author showed that microRNA miR-574-3p and the long ncRNA LINC01003 were correlated with absolute neutrophil counts and hemoglobin values, respectively. Conversely, ACOT9 mRNA was relevant to all complete blood count (CBC) components. two key ncRNAs, miR-574-3p and LINC01003, and one mRNA, ACOT9, associated with innate immune response modulation. In addition, an integrative analysis of post-CCRT ncRNA levels and CBC values revealed that patients with miR-574-3p-LINC01003-ACOT9 values (log2FC) <0 evidenced a better prospect of 30-month disease-specific survival. Therefore, miR-574-3p and LINC01003, which are intricately associated with immune mechanisms and blood cell dynamics, thus suggesting that they serve as potential prognostic biomarkers of ED. Although further biological validations are needed in a future study, I think that the findings of this study are interesting and important. My minor comments are below.

 Comments:

1. There is no information about the machine which is used for measuring CBC.

2. The resolution of each Figure is very poor.

3. The description about the purity of exosome is missing. In addition, the information about sample storage such as period should bed described.  

4. Since this manuscript has one author, “We” should be changed to “I”.

5. The author discussed TLR9 signaling pathway by focusing on red blood cells. As the author mentioned, red blood cells express TLR9. However, TLR9 mainly expressed on plasmacytoid dendritic cells and B cells. How does the author think about the involvement of plasmacytoid dendritic cells and B cells?

Author Response

In the present study, the author performed an integrative analysis of plasma exosomal ncRNAs in conjunction with blood cell dynamics to know their potential prognostic indicators for early death (ED) in concurrent chemo-radiotherapy (CCRT)-treated patients with cervical cancer. The author showed that microRNA miR-574-3p and the long ncRNA LINC01003 were correlated with absolute neutrophil counts and hemoglobin values, respectively. Conversely, ACOT9 mRNA was relevant to all complete blood count (CBC) components. two key ncRNAs, miR-574-3p and LINC01003, and one mRNA, ACOT9, associated with innate immune response modulation. In addition, an integrative analysis of post-CCRT ncRNA levels and CBC values revealed that patients with miR-574-3p-LINC01003-ACOT9 values (log2FC) <0 evidenced a better prospect of 30-month disease-specific survival. Therefore, miR-574-3p and LINC01003, which are intricately associated with immune mechanisms and blood cell dynamics, thus suggesting that they serve as potential prognostic biomarkers of ED. Although further biological validations are needed in a future study, I think that the findings of this study are interesting and important. My minor comments are below.

 Comments:

  1. There is no information about the machine which is used for measuring CBC.

I have added this information in supplementary methods (Supplementary methods, page 2)

  1. The resolution of each Figure is very poor.

All figures have been modified to improve the resolution.

  1. The description about the purity of exosome is missing. In addition, the information about sample storage such as period should bed described.  

I have added an explanation about exosome purity and information about sample storage period in the Supplementary methods. (Supplementary methods, page 3)

  1. Since this manuscript has one author, “We” should be changed to “I”.

I have changed “We” to “I” at all relevant instances.

  1. The author discussed TLR9 signaling pathway by focusing on red blood cells. As the author mentioned, red blood cells express TLR9. However, TLR9 mainly expressed on plasmacytoid dendritic cells and B cells. How does the author think about the involvement of plasmacytoid dendritic cells and B cells?

Disturbance in TLR9-mediated activation of plasmacytoid dendritic cells (pDCs) and B cells may lead to disease progression. However, even if we assume that a portion of 4.1 to 6.1 million/uL RBCs is related to innate immune activation in association with TLR9, the absolute number of RBCs as an immune sensor should have a greater impact than thousands of B cells or fewer pDCs. Therefore, we speculate that the progression of diseases associated with RBC-TLR9 may occur faster and more remarkably.

Round 2

Reviewer 1 Report

Comments and Suggestions for Authors

Dear Oyeon Cho,

I think the manuscript can be accepted in this form. It addresses an emerging aspect in the field.

I hope that you can continue your investigations to improve and validate the results here reported in future studies.

Best regards,

Author Response

Thank you for your positive recommendation.